# Interrater reliability and agreement of the NEUMOBACT checklist about infection-prevention performance of intensive care nurses in simulation-based scenarios

Marta Raurell-Torredà[1,2,3☯], Ignacio Zaragoza-García[2,4,5☯]*, Oscar Arrogante[2,4,6☯], Anna María Aliberch-Raurell[2,7☯], Francisco Javier Sánchez-Chillón[2,8☯], Martín Torralba-Melero[2,9☯], Andrés Rojo-Rojo[2,10☯], Alfonso Muriel-García[11☯], Ana Carolina Amaya Arias[12☯], Juan Roldán-Merino[3,13☯], Mariona Farrés-Tarafa[1,2☯]

1 Department of Fundamental and Clinical Care Nursing, Hospitalet del Llobregat, Universitat de Barcelona, Campus de Bellvitge, Barcelona, Spain, 2 Simulation Group of the Spanish Society for Intensive Care and Coronary Unit Nursing (SEEIUC), Madrid, Spain, 3 Mental Health, Psychosocial and Complex Nursing Care Research Group—2021 SGR 01083, Spain, 4 Department of Nursing, Faculty of Nursing, Physiotherapy and Podology, Universidad Complutense Madrid, Madrid, Spain, 5 Care Research Group (Invecuid), 12 de Octubre Hospital Institute of Health Research (imas12), Madrid, Spain, 6 Research Nursing Group of Instituto de Investigación Sanitaria Gregorio Marañón (IiSGM), Madrid, Spain, 7 Department of Critical Care, Hospital Clínic de Barcelona, Barcelona, Spain, 8 Simulation Centre, Hospital 12 de Octubre, Madrid, Spain, 9 Department of Critical Care, Hospital General Universitario de Albacete, Albacete, Spain, 10 Department of Nursing, Universidad Católica de Murcia, Campus de los Jerónimos, Guadalupe, Murcia, Spain, 11 Biostatistics Unit, Department of Nursing and Physiotherapy, Hospital Universitario Ramón y Cajal, IRYCIS, CIBERESP, Universidad de Alcalá, Madrid, Spain, 12 Division of Non-Communicable Diseases, Ministry of Health and Social Protection, Bogotá, Colombia, 13 Department of Nursing, Sant Joan de Déu Teaching Campus, Universitat de Barcelona, Sant Boi de Llobregat, Barcelona, Spain

☯ These authors contributed equally to this work.
* izaragoz@ucm.es

## Abstract

### Objective

To analyse the interrater reliability of the NEUMOBACT checklist and verify whether consistent results are reproducible.

### Methods

A validation study with a cross-sectional design, compliant with the GRRAS checklist, among ICU nurses attending a SIMULAZERO course with an Objective Structured Clinical Evaluation simulation format, to verify transfer from theory to clinical practice of knowledge and skills in ventilator-associated pneumonia (VAP) and catheter-related bacteraemia (CRB) prevention. A minimum sample size of 111 pairs of nurse raters was calculated. Interrater agreement was analysed using Gwet's AC1 for each item and as a total for each of the three checklists in the NEUMOBACT instrument.

**Data Availability Statement:** All relevant data are within the manuscript and its Supporting Information files.

**Funding:** This project received a Grant for clinical-simulation related research projects from the Spanish Society for Patient Simulation and Safety (Sociedad Española de Simulación Clínica y Seguridad del Paciente, SESSEP). This grant was obtained for the entire research team. The funders had no role in study design, data collection and analysis, decision to publish, or preparation of the manuscript.

**Competing interests:** The authors have declared that no competing interests exist.

## Results

A total of 95 pairs of valid NEUMOBACT checklists were completed by 190 raters with a median age of 29 [25–35] years, 93.7% were female. At the Central Venous Catheter insertion station, Gwet's AC1 was 0.934 (95% CI [0.919–0.949]). Only 2 of the 17 items scored below 0.9. At the Endotracheal Suctioning station, Gwet's AC1 was 0.869 (95% CI [0.851–0.886]). Of the 26 items that made up this station, 16 had an agreement percentage above 0.9, a further 9 were between 0.821 and 0.884, and item 13 had a value of 0.789. At the Patient Care station, Gwet's AC1 was 0.911 (95% CI [0.896–0.927]). Of the 21 items, 17 showed an agreement percentage above 0.9 and 4 were between 0.810 and 0.894.

## Conclusions

The interrater reliability of the NEUMOBACT checklist shows substantial agreement between pairs of raters and is therefore validated in this large sample of ICU nurses.

## Relevance to clinical practice

The NEUMOBACT checklist can be useful for assessing skills before and after training in VAP and CRB prevention measures and during debriefing (post-simulation feedback), to reinforce the scientific evidence behind actions and decisions for items that have been performed incorrectly, thus consolidating training already received.

## Introduction

Intensive care units (ICUs) underwent structural and organisational changes as a result of the Covid-19 pandemic, and these changes lead to a two- to threefold increase in the incidence rates of ventilator-associated pneumonia (VAP) and catheter-related bacteraemia (CRB), together with increased length of lCU stays and intra-ICU mortality [1]. Measures to address these problems included a proposal by the Ministry of Health's board of experts to "promote Zero Project training among existing, newly recruited and temporary ICU healthcare providers". These projects include Bacteraemia Zero and Pneumonia Zero, which are training and epidemiological surveillance programmes designed to reduce the incidence of CRB and VAP, respectively, that are implemented across Spain and are supported by experts from the Spanish Society for Intensive and Critical Care Medicine and Coronary Units (SEMICYUC) and the Spanish Society for Intensive Care and Coronary Unit Nursing (SEEIUC).

The SEEIUC Simulation working group runs a training course called SIMULAZERO, which applies teaching through simulation to assess ICU nurses' knowledge and skills in VAP and CRB prevention [2]. The SIMULAZERO course is based on an Objective Structured Clinical Evaluation (OSCE) composed of two training scenarios for technical skills (endotracheal suctioning and central venous catheter insertion, respectively) and a training scenario for non-technical skills [3], which, in this case, concerns decision-making, resource management, safe practice and reduction of adverse events related to the care of critically ill patients.

In simulation, OSCEs are considered the gold standard for competency-based assessments. In OSCEs, candidates rotate through a series of time-limited stations in a circuit designed to assess professional performance in a simulated environment. These assessments facilitate the objective evaluation of a complex range of skills, knowledge, and attitudes through direct

observation [4], while providing an adequate method for engaging and integrating theory with practice [5]. These evaluations can be training or assessment based, or both [6]. Various checklists are used in the ICU to guide evidence-based clinical practice, with a compliance rate ranging from 73% to 91%, improving care outcomes [7], but no checklist was found in the literature to assess ICU nurses' performance during an OSCE, and therefore the NEUMOBACT checklist was created, designed and face validity and content [8]. The aim of this study was to analyse the interrater reliability of the NEUMOBACT checklist and verify whether consistent results are reproducible.

## Materials and methods

A descriptive study of a metric nature, reported using the GRRAS checklist (S1 Checklist) for reporting of studies of reliability and agreement [9] and was conducted among ICU nurses attending the SIMULAZERO course at the 47th National Congress of SEEIUC. Recruitment was conducted by convenience from 13 to 15 June 2022. The nurses worked at various ICUs across Spain, permitting the identification of potential heterogeneity among those ICUs. Nurses were included if they had participated in the course as raters, completed all the NEUMOBACT checklist items, and agreed to participate in the study.

### Sample calculation

According to the COSMIN Study Design checklist for patient-reported outcome measurement instruments [10], a minimum of 100 pairs of assessments by at least two unbiased evaluators is required to analyze inter-observer reliability. Two independent observers should be used to assess the simulation stations under similar environmental conditions, having received the same instructions. To account for expected losses, a further 10% was added, resulting in a sample of 111 pairs of nurses to evaluate the NEUMOBACT checklist during the simulation.

### Variables and information sources

The data were collected during the SIMULAZERO course, based on a multi-station OSCE-format simulation (specifically 3), whereby participants rotated around a circuit of sequential stations. Each station presented a clinical case, requiring students to perform a specific set of tasks by applying a range of knowledge, skills and attitudes, resulting in the simultaneous assessment of several competence-based components. Each participant started their test at a different station, but at the same time as each other (about 10 minutes). The scenarios were of a similar length so that participants would leave one station and start the next in synchronisation. The resources used included standard materials commonly available in intensive care units, as well as basic CPR manikins, avoiding any material limitations for future replication in other ICUs. Two simultaneous rounds were conducted, with each station requiring one technician, two assessors, and two OSCE coordinators.

Raters received specific training in their roles as raters to provide them with detailed assessment criteria and the meaning of each item. The raters were paired up and given the NEUMOBACT checklist, which had been previously developed by experts using the Delphi technique, to use for each station assessment, with only two response options for each item: correct/incorrect (Table 1). They were given 10 minutes to familiarise themselves with the checklist and were reminded of the importance of completing all the items when assessing each station. They were also instructed not to share the information with each other and for this purpose they were positioned one on each side of the simulator. Since the three NEUMOBACT checklists are independent of each other, each rater pair started the assessment at a different station and followed the circuit described above. The rater pairs were also asked to respond to

**Table 1. NEUMOBACT-English version.**

| Central Venous Catheter (CVC) Insertion | | Correct | Incorrect |
|---|---|---|---|
| 1 | Select the insertion site by weighing up the risk of infection against the risk of mechanical complications. | | |
| 2 | Use subclavian vein access unless contraindicated (anatomical deformities, coagulation disorders, kidney disease that might require dialysis). | | |
| 3 | If jugular vein access is selected, use the right side to reduce non-infectious complications (unless ultrasound-guided insertion is performed). | | |
| 4 | Always use catheters with the smallest number of lumens possible. | | |
| 5 | When inserting a thoracic CVC, the patient should remain in the Trendelenburg position if necessary and if not contraindicated. | | |
| 6 | Prior to skin asepsis, clean the insertion site with chlorhexidine soap and water, then rinse and allow to dry fully. | | |
| 7 | Perform hand hygiene with soap and water (40–60 seconds) or alcohol solution (20–30 seconds). | | |
| 8 | For skin asepsis prior to catheter insertion, a 0.5–2% alcohol-based chlorhexidine solution should preferably be used. Use 70˚ alcohol or povidone-iodine only in the case of hypersensitivity to chlorhexidine. The antiseptic must be completely dry before catheter insertion (if povidone iodine is used, at least two minutes drying time). | | |
| 9 | Do NOT palpate the puncture site after applying the antiseptic, unless an aseptic technique is used. | | |
| 10 | Use maximum barrier measures (mask, cap, eye protection and gown, sterile drapes, sheets and gloves) for the CVC insertion. | | |
| 11 | Insertion assistants must comply with the above measures. They must don a cap and mask, at a minimum. | | |
| 12 | The sterile field must cover the patient's entire body. | | |
| 13 | Before connecting any components to a catheter lumen, aspirate blood from the patient through each lumen to prevent air from entering the blood stream. | | |
| 14 | Apply a sterile dressing to the catheter insertion site before the barrier measures are removed. | | |
| 15 | Use a transparent, semi-permeable or chlorhexidine-impregnated dressing to cover the insertion site. A gauze dressing may be used if there is bleeding. | | |
| 16 | Note the catheter insertion date in the nursing records and on the dressing. | | |
| 17 | Place needle-free connectors only at sites where boluses are to be administered. | | |
| **Endotracheal suctioning (ETS)** | | Correct | Incorrect |
| 1 | Regardless of whether or not the patient is sedated, inform them of the technique to be performed | | |
| 2 | Perform hand hygiene with soap and water (40–60 seconds) or alcohol solution (20–30 seconds). | | |
| 3 | Don non-sterile gloves. | | |
| 4 | Don personal protective equipment: mask and eye protection or mask with face shield. | | |
| 5 | Do not instil normal saline routinely. Normal saline instillation may increase secretions; use is recommended as an exception only (when secretions need to be thinned or if there is a tendency for plug formation). | | |
| 6 | Hyperoxygenate the patient before and after suctioning. | | |
| 7 | Choose to use the ventilator to hyperoxygenate/hyperventilate. It is more recommendable to use the ventilator than a bag valve mask (Ambu®). | | |
| 8 | Activate the ventilator setting that hyperoxygenates/hyperinflates the patient. | | |
| 9 | Consider closed-system suctioning when $PaO_2/FiO_2$ is <200 and PEEP levels are set high. | | |
| 10 | **Suction technique: OPEN-SUCTION SYSTEM** | | |
| 10.1 | Select a suction pressure between 100–150 mmHg | | |
| 10.2 | Put a single-use sterile glove on the dominant hand, which will hold the suction catheter. | | |
| 10.3 | Prevent micro-atelectasis: use a catheter with an appropriate diameter (half the internal diameter of the ETT). | | |
| 10.4 | Sterile single-use catheter–insert the catheter into the bronchial tree without suctioning and then suction for no more than 15 seconds at a time. Maximum 3 times. | | |
| 10.5 | After suctioning, wash out the tube to the suction unit with sterile water | | |
| 11 | **Suction technique: CLOSED-SUCTION SYSTEM** | | |
| 11.1 | The first time the technique is performed, prepare the system: choose an appropriate-sized closed-suction catheter considering the ETT diameter. | | |
| 11.2 | Connect the elbow/T-piece between the ventilator and the ETT connection. | | |
| 11.3 | Place the sticker showing the date the device was connected in a visible place. | | |
| 11.4 | Connect the suction system to the closed-suction catheter. Select a suction unit pressure between 100–150 mmHg | | |
| 11.5 | Open the valve system to the ETT and insert the catheter without applying suction to about 2 cm above the carina. | | |
| 11.6 | Press the suction button for less than 15 seconds. | | |

*(Continued)*

**Table 1.** (Continued)

| Central Venous Catheter (CVC) Insertion | | Correct | Incorrect |
|---|---|---|---|
| 11.7 | Withdraw the catheter slowly while continuing to apply suction until you reach the guide mark. | | |
| 11.8 | Close the valve system between the catheter and the ETT. | | |
| 11.9 | Clean the suction catheter: connect a 20 ml syringe with sterile water to the washout port. Press the suction button on the catheter and observe how the syringe empties and the catheter is washed out. | | |
| 11.10 | Close the suction system | | |
| | **Items applicable to both suction techniques** | | |
| 12 | Check the cuff pressure after suctioning (and at least once per shift). Pressure should be maintained between 20 and 30 $cmH_2O$ to prevent subglottic secretions moving into the lower airway and to avoid vascular compromise of the trachea. Continuous cuff pressure measurement systems are recommended. | | |
| 13 | Perform hand hygiene with soap and water (40–60 seconds) or alcohol solution (20–30 seconds). | | |

| Patient care (PC) | | Correct | Incorrect |
|---|---|---|---|
| **Event no. 1: Patient with cough.** Consider whether suctioning is indicated by performing: | | | |
| 1a | Auscultation of breath sounds during the expiratory phase of the respiratory cycle (with a stethoscope over the trachea, in the sternal area). High specificity/sensitivity | | |
| 1b | Changes in the flow-volume curve shape of the ventilator monitor (accelerations/decelerations from baseline). High specificity/sensitivity | | |
| 1c | Decrease in saturation. Low specificity because desaturation can occur in situations other than secretion retention, such patient-ventilator mismatch, patient mobilisation and bronchospasm. | | |
| 1d | Increase in peak pressure in volume-controlled systems. Low specificity because desaturation can occur in situations other than secretion retention, such patient-ventilator mismatch, patient mobilisation and bronchospasm. | | |
| 1e | Decrease in tidal volume in pressure-controlled systems. Low specificity because desaturation can occur in situations other than secretion retention, such patient-ventilator mismatch, patient mobilisation and bronchospasm. | | |
| **Event no. 2: Oral hygiene** | | | |
| 2 | Before performing oral hygiene, check that cuff pressure > 20 $cmH_2O$. | | |
| 3 | Keep the head of the bed raised to perform oral hygiene. | | |
| 4 | Use 0.12–0.2% chlorhexidine to perform oral hygiene. | | |
| **Event no. 3: Catheter change** | | | |
| 5 | Always use catheters with as few lumens as possible. | | |
| 6 | Catheters should not be changed over a guidewire. | | |
| 7 | Only in patients with suspected bacteraemia and limited venous access, the catheter may be changed over a guidewire, always sending the catheter tip for culture. | | |
| 8 | If possible, in the case of multi-lumen catheters, select and designate one lumen for lipid emulsions only (parenteral nutrition, propofol). | | |
| 9 | All catheter replacements and manoeuvres must be recorded. | | |
| **Event no. 4: Changing giving sets and connections** | | | |
| 10 | Change giving sets between days 4 and 7, unless hubs look dirty or have been accidentally disconnected. | | |
| 11 | Do not use antibiotic or antiseptic ointments to protect the insertion site. | | |
| 12 | Use sterile gloves for dressing changes (one pair of gloves for each dressing). | | |
| 13 | Note the catheter insertion date in the nursing records and on the dressing. | | |
| 14 | Wash hands and don non-sterile gloves before handling equipment, connections and valves. | | |
| 15 | Place needle-free connectors only at sites where boluses are to be administered. Needle-free bungs protect staff, but can pose a risk of infection if not used correctly. | | |
| 16 | Use as few three-way taps as possible and remove them when they are not essential. | | |
| 17 | Clean injection caps with alcohol-based chlorhexidine before accessing the system. | | |

sociodemographic questions on their age, gender, nursing experience, ICU nursing experience and whether they had postgraduate training.

S1 File describes the learning objective, participants' roles and actors for each of the three stations in the circuit.

## Data analysis

Qualitative variables were expressed as frequency and percentage and quantitative variables as mean and standard deviation or median and interquartile range (P25-P75), depending on the data distribution. To analyse the reliability of the instrument, interrater agreement was calculated using Gwet's AC1 for each NEUMOBACT checklist item, and for each station as a total value. Gwet's AC1 coefficient method was used because it is more stable and less affected by prevalence and marginal probability than Cohen's Kappa [11, 12]. Gwet's AC1 values range from -1 to 1, where a negative value indicates significant disagreement, zero indicates what would be expected by chance, values ≤0.3 indicate slight agreement, 0.31–0.6 moderate agreement, and ≥0.7 substantial agreement [11].

## Ethical aspects

The project was approved by the Bioethics Commission of the University of Barcelona (code number: IRB00003099). The investigators undertook to comply with Organic Law 3/2018 of 5 December on personal data protection and the guarantee of digital rights. The participants were informed about the study's purpose and invited to participate. Consent was given verbally and confirmed again verbally when participants chose to respond to the checklist. In order to enhance data protection, only the principal investigator recorded and had access to the socio-demographic and academic data of each rater.

## Results

### Demographic characteristics

95 pairs of valid NEUMOBACT checklists were collected. The raters (n = 190) had a median age (median [P25-P75]) of 29 [25–35] years and 93.7% were female. They had 5 [2–8] years of nursing experience and 3 [2–7] years of ICU-specific experience. A total of 54.7% reported postgraduate-level education.

### Reliability

At the Central Venous Catheter (CVC) insertion station, the total value for Gwet's AC1 was 0.934 (95% CI [0.919–0.949]). All items had an agreement percentage above 0.9 except item 10 ("*Use maximum barrier measures (mask, cap, eye protection and gown, sterile drapes, sheets and gloves) for CVC insertion*") and item 17 ("*Place needle-free connectors only at sites where boluses are to be administered*") (Table 2).

At the Endotracheal Suctioning (ETS) station, the total value for Gwet's AC1 was 0.869 (95% CI [0.851–0.886]). Of the 26 items assessed at this station, 16 showed agreement percentages of >0.9, a further 9 items (specifically, items 4, 9, 10.1, 10.3, 10.5, 11.3, 11.4, 11.8 and 12) had percentages between 0.821 and 0.884, and item 13 had a value of 0.789 (Table 2).

At the Patient Care (PC) station, the total value for Gwet's AC1 was 0.911 (95% CI [0.896–0.927]). Of the 21 items assessed at this station, 17 showed agreement percentages of >0.9, and the other 4 items (specifically, items 1e, 7, 14 and 17) had percentages between 0.810 and 0.894 (Table 2).

To improve the reliability of items with agreement percentages below 0.9, the investigators, through a focus group discussion, reviewed the checklist and proposed the following changes to make it easier for raters to understand the items. For item 10 of the CVC insertion station, the word "*ALL*" was added to clarify that the item should be marked as correct when all the measures are applied and as incorrect when only some of them are applied. The content of item 17 was expanded, adding "*intermittent medication and/or bolus*". For the ETS station, the

**Table 2. Agreement and Gwet's AC1 for the NEUMOBACT checklist.**

| Item | Agreement (%) | Gwet's AC1 | 95% CI |
|---|---|---|---|
| Central Venous Catheter (CVC) | | | |
| 1 | 0.957 | 0.947 | 0.893; 1.000 |
| 2 | 0.936 | 0.908 | 0.833; 0.984 |
| 3 | 0.989 | 0.988 | 0.964; 1.000 |
| 4 | 0.926 | 0.894 | 0.813; 0.975 |
| 5 | 0.936 | 0.874 | 0.775; 0.973 |
| 6 | 0.936 | 0.914 | 0.842; 0.985 |
| 7 | 100 | - | - |
| 8 | 0.968 | 0.965 | 0.924; 1.000 |
| 9 | 0.978 | 0.975 | 0.939; 1.000 |
| 10 | 0.863 | 0.836 | 0.741; 0.931 |
| 11 | 0.968 | 0.940 | 0.871; 1.000 |
| 12 | 0.957 | 0.956 | 0.911; 1.000 |
| 13 | 100 | - | - |
| 14 | 0.957 | 0.951 | 0.901; 1.000 |
| 15 | 0.968 | 0.966 | 0.927; 1.000 |
| 16 | 0.957 | 0.927 | 0.856; 0.999 |
| 17 | 0.873 | 0.793 | 0.673; 0.913 |
| **TOTAL** | 0.951 | 0.934 | 0.919; 0.949 |
| Endotracheal Suctioning (ETS) | | | |
| 1 | 0.905 | 0.814 | 0.696; 0.932 |
| 2 | 0.926 | 0.884 | 0.796; 0.972 |
| 3 | 0.957 | 0.950 | 0.899; 1.000 |
| 4 | 0.894 | 0.794 | 0.670; 0.918 |
| 5 | 0.947 | 0.939 | 0.884; 0.994 |
| 6 | 0.915 | 0.887 | 0.805; 0.969 |
| 7 | 0.926 | 0.910 | 0.840; 0.980 |
| 8 | 0.968 | 0.963 | 0.921; 1.000 |
| 9 | 0.884 | 0.786 | 0.662; 0.911 |
| 10.1 | 0.873 | 0.777 | 0.651; 0.904 |
| 10.2 | 100 | - | - |
| 10.3 | 0.863 | 0.815 | 0.708; 0.922 |
| 10.4 | 0.957 | 0.954 | 0.907; 1.000 |
| 10.5 | 0.884 | 0.831 | 0.726; 0.935 |
| 11.1 | 0.905 | 0.877 | 0.792; 0.961 |
| 11.2 | 0.947 | 0.940 | 0.886; 0.994 |
| 11.3 | 0.821 | 0.652 | 0.496; 0.807 |
| 11.4 | 0.863 | 0.764 | 0.634; 0.893 |
| 11.5 | 0.968 | 0.961 | 0.916; 1.000 |
| 11.6 | 0.957 | 0.955 | 0.909; 1.000 |
| 11.7 | 0.926 | 0.919 | 0.855; 0.982 |
| 11.8 | 0.852 | 0.807 | 0.699; 0.915 |
| 11.9 | 0.905 | 0.874 | 0.788; 0.960 |
| 11.10 | 0.957 | 0.953 | 0.905; 1.000 |
| 12 | 0.873 | 0.768 | 0.639; 0.898 |
| 13 | 0.789 | 0.645 | 0.487; 0.802 |
| **TOTAL** | 0.910 | 0.869 | 0.851; 0.886 |

(*Continued*)

**Table 2.** (Continued)

| Item | Agreement (%) | Gwet's AC1 | 95% CI |
|---|---|---|---|
| | Patient Care (PC) | | |
| 1a | 0.947 | 0.902 | 0.815; 0.988 |
| 1b | 0.968 | 0.960 | 0.914; 1.000 |
| 1c | 0.957 | 0.955 | 0.909; 1.000 |
| 1d | 0.978 | 0.978 | 0.946; 1.000 |
| 1e | 0.831 | 0.711 | 0.569; 0.854 |
| 2 | 0.989 | 0.984 | 0.954; 1.000 |
| 3 | 0.957 | 0.949 | 0.897; 1.000 |
| 4 | 100 | - | - |
| 5 | 0.926 | 0.920 | 0.858; 0.982 |
| 6 | 0.978 | 0.975 | 0.940; 1.000 |
| 7 | 0.810 | 0.671 | 0.518; 0.823 |
| 8 | 0.947 | 0.944 | 0.893; 0.995 |
| 9 | 0.905 | 0.856 | 0.759; 0.953 |
| 10 | 0.915 | 0.883 | 0.798; 0.967 |
| 11 | 0.947 | 0.937 | 0.879; 0.994 |
| 12 | 0.968 | 0.963 | 0.919; 1.000 |
| 13 | 0.957 | 0.952 | 0.903; 1.000 |
| 14 | 0.894 | 0.872 | 0.789; 0.956 |
| 15 | 0.947 | 0.930 | 0.867; 0.993 |
| 16 | 0.905 | 0.884 | 0.804; 0.964 |
| 17 | 0.852 | 0.829 | 0.733; 0.925 |
| **TOTAL** | 0.932 | 0.911 | 0.896; 0.927 |

word "ALL" was added to item 4 for the same reason as for the CVC station item. For the PC station, it was proposed to make the word "and" uppercase and bold in items 13 and 14 to highlight that both points in the item have to be performed (Table 3).

## Discussion

Very good interrater reliability was achieved on all items of the NEUMOBACT checklist for use in the assessment of ICU nurses' knowledge and skills in VAP and CRB prevention using

**Table 3. Changes proposed to improve interrater agreement.**

| Central Venous Catheter (CVC) Insertion | |
|---|---|
| 10. Use maximum barrier measures (mask, cap, eye protection and gown, sterile drapes, sheets and gloves) for the CVC insertion | 10. Use ALL **maximum barrier measures** (mask, cap, eye protection and gown, sterile drapes, sheets and gloves) for the CVC insertion |
| 17. Place needle-free connectors only at sites where boluses are to be administered | 17. Place needle-free connectors only at sites where intermittent medication and/or bolus are to be administered |
| **Endotracheal Suctioning (ETS)** | |
| 4. Don personal protective equipment: mask and eye protection or mask with face shield | 4. Don ALL personal protective equipment: mask and eye protection or mask with face shield |
| **Patient Care (PC)** | |
| 13. Note the catheter insertion date in the nursing records and on the dressing. | 13. Note the catheter insertion date in the nursing records AND on the dressing |
| 14. Wash hands and don non-sterile gloves before handling equipment, connections and valves. | 14. Wash hands AND don non-sterile gloves before handling equipment, connections and valves. |

simulation-based training. The checklist can be useful to assess skills before and after training in the Pneumonia Zero and Bacteraemia Zero project, to verify theory-to-practice transfer, which is a controversial aspect of training [13, 14] and, during debriefing (post-simulation feedback), to reinforce the scientific evidence behind actions and decisions for items that have been performed incorrectly [15–18]. Although this program has been conducted with ICU nursing professionals, it could be implemented in undergraduate studies, as various authors highlight the benefits of using simulation methodology during the pre-professional stage [19, 20].

Mogyoródi et al. [21] analysed before-and-after compliance with VAP prevention measures in clinical practice with a PowerPoint training programme and found a significant improvement in compliance three months after training, but a decrease to baseline at 12 months. Jansson et al. [22] found no impact on skills following simulation training on adherence to oral care recommendations. Gerolemou et al. [23] evaluated simulation-based training for CVC insertion and demonstrated an 85% reduction in the incidence of CRB infections in a critical care unit after the intervention.

Various VAP and CRB prevention checklists and care bundles have been published in the literature. These instruments were either created to assess compliance with pneumonia prevention measures in clinical practice [24–26] or designed to be applied during simulation of CVC-related skills [23, 27], but as far as we are aware, our NEUMOBACT checklist is the only one that contains all the necessary items for VAP and CRB prevention, created and validated by infection prevention and simulation experts [9].

Furthermore, unlike other instruments, the NEUMOBACT checklist requires no prior training or preparation for use during the simulation [28, 29]. This study, with a large sample of 190 raters, shows that if clear instructions are giving on using the checklist, (i.e., mark each item as correct/incorrect when those actions/decisions are observed during the simulation), it is feasible to achieve good interrater agreement for checklist completion.

## Limitations

In the sample analysed, half of the nurses had postgraduate education. This level of education may have enhanced the nurses' understanding of the checklist items compared to other nurses with undergraduate training only, because they are more familiar with complex ICU procedures, the theory behind those procedures, and simulation scenarios in general, as it is a widely used teaching method in postgraduate training [30]. To compensate for this possible bias, the raters were informed that the checklists used during the SIMULAZERO course were for research to assess interrater reliability, and that at no time would the checklists be analysed to assess participant performance of actions/decisions during the simulation.

It is essential to have validated rubrics in Spain to evaluate simulation-based activities [31]. Since developing an instrument is both costly and time-consuming, adapting existing instruments into another language offers several advantages. On one hand, it reduces research costs while preserving the psychometric properties of the original instrument. On the other hand, it allows for comparison of equally valid and reliable results with those from other national and international studies that have used the same instrument [32].

## Conclusions

The interrater reliability of the NEUMOBACT checklist shows substantial agreement between pairs of raters and is therefore validated in this large sample of ICU nurses. The checklist requires no prior training and is validated and reliable for assessing nurses' performance during simulation-based training in VAP and CRB preventive measures.

## Supporting information

**S1 Checklist. Checklist for reporting of studies of reliability and agreement (GRRAS checklist).**
(PDF)

**S1 File. Description of the simulation stations used in the OSCE-format SIMULAZERO course.**
(DOCX)

## Acknowledgments

The authors are grateful to the nurses who participated in the workshops at the Seville congress and to Maria Pérez Riart and Purificación Pérez Teran from the Hospital del Mar for their teaching and methodological support.

## Author Contributions

**Conceptualization:** Marta Raurell-Torredà, Ignacio Zaragoza-García, Oscar Arrogante, Anna María Aliberch-Raurell, Francisco Javier Sánchez-Chillón, Martín Torralba-Melero, Andrés Rojo-Rojo, Alfonso Muriel-García, Ana Carolina Amaya Arias, Juan Roldán-Merino, Mariona Farrés-Tarafa.

**Data curation:** Marta Raurell-Torredà, Oscar Arrogante, Anna María Aliberch-Raurell, Francisco Javier Sánchez-Chillón, Martín Torralba-Melero, Andrés Rojo-Rojo, Alfonso Muriel-García, Juan Roldán-Merino, Mariona Farrés-Tarafa.

**Formal analysis:** Marta Raurell-Torredà, Ignacio Zaragoza-García, Oscar Arrogante, Alfonso Muriel-García, Ana Carolina Amaya Arias, Juan Roldán-Merino, Mariona Farrés-Tarafa.

**Funding acquisition:** Marta Raurell-Torredà, Francisco Javier Sánchez-Chillón, Martín Torralba-Melero, Andrés Rojo-Rojo, Mariona Farrés-Tarafa.

**Investigation:** Marta Raurell-Torredà, Ignacio Zaragoza-García, Oscar Arrogante, Anna María Aliberch-Raurell, Francisco Javier Sánchez-Chillón, Martín Torralba-Melero, Andrés Rojo-Rojo, Mariona Farrés-Tarafa.

**Methodology:** Marta Raurell-Torredà, Ignacio Zaragoza-García, Anna María Aliberch-Raurell, Francisco Javier Sánchez-Chillón, Martín Torralba-Melero, Andrés Rojo-Rojo, Alfonso Muriel-García, Ana Carolina Amaya Arias, Juan Roldán-Merino, Mariona Farrés-Tarafa.

**Project administration:** Marta Raurell-Torredà, Anna María Aliberch-Raurell.

**Supervision:** Marta Raurell-Torredà, Mariona Farrés-Tarafa.

**Validation:** Oscar Arrogante.

**Writing – original draft:** Marta Raurell-Torredà, Ignacio Zaragoza-García, Oscar Arrogante, Mariona Farrés-Tarafa.

**Writing – review & editing:** Marta Raurell-Torredà, Ignacio Zaragoza-García, Oscar Arrogante, Anna María Aliberch-Raurell, Francisco Javier Sánchez-Chillón, Martín Torralba-Melero, Andrés Rojo-Rojo, Alfonso Muriel-García, Ana Carolina Amaya Arias, Juan Roldán-Merino, Mariona Farrés-Tarafa.

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
