## [Decision Letter · Decision Letter 0]

27 Sep 2024

PONE-D-24-06321Interrater reliability and agreement of the NEUMOBACT checklist about infection-prevention performance of intensive care nurses in simulation-based scenarios.PLOS ONE

Dear Dr. Zaragoza-García,

Thank you for submitting your manuscript to PLOS ONE. After careful consideration, we feel that it has merit but does not fully meet PLOS ONE’s publication criteria as it currently stands. **Therefore, we invite you to submit a revised version of the manuscript that addresses the points raised during the review process.**

We look forward to receiving your revised manuscript.

Kind regards,

Javier Fagundo-Rivera, PhD

Academic Editor

PLOS ONE

Journal Requirements:

3. We note that your Data Availability Statement is currently as follows: “All relevant data are within the manuscript and in Supporting Information files.”

Please confirm at this time whether or not your submission contains all raw data required to replicate the results of your study. Authors must share the “minimal data set” for their submission. PLOS defines the minimal data set to consist of the data required to replicate all study findings reported in the article, as well as related metadata and methods (https://journals.plos.org/plosone/s/data-availability#loc-minimal-data-set-definition). For example, authors should submit the following data: - The values behind the means, standard deviations and other measures reported; - The values used to build graphs; - The points extracted from images for analysis. Authors do not need to submit their entire data set if only a portion of the data was used in the reported study. If your submission does not contain these data, please either upload them as Supporting Information files or deposit them to a stable, public repository and provide us with the relevant URLs, DOIs, or accession numbers. For a list of recommended repositories, please see https://journals.plos.org/plosone/s/recommended-repositories. If there are ethical or legal restrictions on sharing a de-identified data set, please explain them in detail (e.g., data contain potentially sensitive information, data are owned by a third-party organization, etc.) and who has imposed them (e.g., an ethics committee). Please also provide contact information for a data access committee, ethics committee, or other institutional body to which data requests may be sent. If data are owned by a third party, please indicate how others may request data access.

**Additional Editor Comments**:

Dear Authors,

Two reviewers have completed their evalation of your manuscript.

Look carefully at their comments and complete your review with the required modifications.

We apologize for the delay in reaching to a decision, and we hope to continue processing your manuscript after minor revisions.

**Congratulations for your work and good luck.**

**Reviewers' comments:**

Reviewer's Responses to Questions

**Comments to the Author**

1. Is the manuscript technically sound, and do the data support the conclusions?

Reviewer #1: Yes

Reviewer #2: Yes

2. Has the statistical analysis been performed appropriately and rigorously? 

Reviewer #1: Yes

Reviewer #2: Yes

3. Have the authors made all data underlying the findings in their manuscript fully available?

Reviewer #1: No

Reviewer #2: Yes

4. Is the manuscript presented in an intelligible fashion and written in standard English?

Reviewer #1: Yes

Reviewer #2: Yes

5. Review Comments to the Author

Reviewer #1: I would like to congratulate the authors of the manuscript for their work, as I believe that both its subject matter and execution represent a significant contribution to the nursing scientific community, both in clinical and research areas. Therefore, we now make a series of minor observations to ensure that the study meets the quality standards of PLOS ONE.

In general terms, I recommend that the authors review the writing style of the manuscript to ensure compliance with the spelling recommendations in scientific publications.

Introduction

(Lines 109-113) I believe it is necessary to include some information regarding the typical structure of OSCEs, such as the number of scenarios, recommended duration, the context in which they are usually conducted, evaluation forms, etc. The authors recommend that they consult Harden's original reference for this purpose.

(Lines 114-116) I think it is relevant to describe what is already known about reliability and agreement in the checklists used in the intensive care unit, to contextualize the need to conduct one concerning an OSCE in this setting.

Methodology

I would recommend that the authors include a figure based on a flow chart detailing each stage of the study.

(Lines 145-147) The authors mention that the recruitment was carried out at a conference, suggesting convenience sampling. However, this is not explicitly mentioned, which I believe would add transparency to the article.

(Lines 153-159) In addition to mentioning the COSMIN checklist, I think it is important to mention the statistical parameters used and the value at which they were set to calculate the minimum sample size required for the study.

(Lines 161-168) I would further detail the most relevant characteristics of the simulation scenarios (duration, simulated clinical situation, necessary material and human resources, characteristics of the simulators, etc.).

(Lines 170-172) It is necessary to specify the methodology followed for developing the initial version of NEUMOBACT (it is not mentioned whether it was a literature review, focus groups with thematic analysis and triangulation, etc.).

(Lines 202-209) It would be important for the authors to provide the DOI or registration ID of the study in registries such as the Open Science Framework.

Results

(Lines 235-237) I believe it is relevant to detail the methodology followed for developing the modified version of NEUMOBACT (it is not mentioned whether it was a literature review, focus groups with thematic analysis and triangulation, etc.).

Discussion

(Line 278) I would mention future perspectives for upcoming research related to the topic. I would also highlight the potential benefit of applying simulated learning methodologies at the undergraduate and postgraduate levels, such as MAES or OSCEs with validated checklists, to improve patient safety and quality of care.

Limitations

(Lines 280-288) I believe it is important to incorporate more citations to support the claims made by the authors. I also think they should mention the geographical limitation of NEUMOBACT validation, as well as possible recommendations for its use in contexts outside of Spanish-speaking ones.

References

From reference number 10 onward, there is an indentation that is not present in references 1-9, so I recommend that the authors review the format used.

Reviewer #2: I would like to congratulate the authors of the manuscript for their work, as I believe that the topic is timely and will probably be of interest to the readers of the journal. Therefore, we now make a series of minor reviews in different sections to ensure that the study meets the quality standards of PLOS ONE:

Introduction

(Lines 109-112) It is true that OSCEs are considered very useful for simulation. However, in this sense, for competence-based assessments for nurses we also find other methodologies that are widespread in Spain such as the MAES©, which are also used in university training. It would be interesting to develop this idea in greater depth with references such as:

- Romero-Castillo R, Garrido-Bueno M, Fernández-León P. Nursing students' perceptions and satisfaction with a self-learning methodology in simulated environments: A mixed-methods study. Nurse Educ Pract. 2024 Sep 12;80:104141. doi: 10.1016/j.nepr.2024.104141.

- Díaz Agea JL, Ramos-Morcillo AJ, Amo Setien FJ, Ruzafa-Martínez M, Hueso-Montoro C, Leal-Costa C. Perceptions about the Self-Learning Methodology in Simulated Environments in Nursing Students: A Mixed Study. Int J Environ Res Public Health. 2019 Nov 22;16(23):4646. doi: 10.3390/ijerph16234646.

Methodology

(Lines 145-146) In addition to mentioning the GRRAS checklist, I think it is important to add it completed as supplementary material.

(Lines 161-168) I would further detail the most relevant characteristics of the simulation scenarios (duration, situations, material and human resources...).

Discussion

(Lines 252-258) I would mention future perspectives for upcoming research related to the topic. For example, I would also highlight the potential benefit of applying simulated learning methodologies at the undergraduate levels. It would be interesting to develop this idea in greater depth with references such as:

- Romero-Castillo R, Garrido-Bueno M, Fernández-León P. Nursing students' perceptions and satisfaction with a self-learning methodology in simulated environments: A mixed-methods study. Nurse Educ Pract. 2024 Sep 12;80:104141. doi: 10.1016/j.nepr.2024.104141.

- Díaz Agea JL, Ramos-Morcillo AJ, Amo Setien FJ, Ruzafa-Martínez M, Hueso-Montoro C, Leal-Costa C. Perceptions about the Self-Learning Methodology in Simulated Environments in Nursing Students: A Mixed Study. Int J Environ Res Public Health. 2019 Nov 22;16(23):4646. doi: 10.3390/ijerph16234646.

Limitations

(Lines 280-288) Is it only the level of education of the nurses as a limitation? I think others limitations can be mentioned such as geographical limitation or those typical of cross-sectional designs.

References

I recommend to review this section in greater depth (format and content). An indentation appears starting from reference 10 (included). Additionally, some references (8, 9…) are not complete.

6. PLOS authors have the option to publish the peer review history of their article (what does this mean?). If published, this will include your full peer review and any attached files.

Reviewer #1: No

Reviewer #2: No

---

## [Author Response · Author response to Decision Letter 0]

4 Oct 2024

Authors: Thank you for the opportunity to review the manuscript. We have now addressed the editor's and reviewers' comments. Changes made to the manuscript are highlighted in red.

Journal Requirements:

Authors: Thank you for the feedback. We have revised the document and adapted it to PLOS ONE standards.

Authors: Thank you for your comment. In the Competing Interests section where the ‘Financial Disclosures’ are declared, we stated that we had no conflict of interest. The funding received was from a not-for-profit company. The funders had no role in study design, data collection and analysis, decision to publish, or preparation of the manuscript.

On the other hand, it should be noted that this information has only been included in the ‘Funding Information’ section as there is no ‘Financial Disclosures’ section and the PloOne rules specify that: ‘Enter this statement in the Financial Disclosure section of the submission form. Do not include it in your manuscript file’.

The information requested by the magazine is attached below. We have added the web address:

• Specific grant numbers: there is no specific number, only the year of concession.

• Initials of authors who received each award: we have reflected that the grant was awarded equally to all authors.

• Full names of commercial companies that funded the study or authors: There are no commercial enterprises related to the grant

• Initials of authors who received salary or other funding from commercial companies: does not apply to our scholarship

• URLs to sponsors’ websites: we have added it

Please confirm at this time whether or not your submission contains all raw data required to replicate the results of your study. Authors must share the “minimal data set” for their submission. PLOS defines the minimal data set to consist of the data required to replicate all study findings reported in the article, as well as related metadata and methods (https://journals.plos.org/plosone/s/data-availability#loc-minimal-data-set-definition). For example, authors should submit the following data: - The values behind the means, standard deviations and other measures reported; - The values used to build graphs; - The points extracted from images for analysis. Authors do not need to submit their entire data set if only a portion of the data was used in the reported study. If your submission does not contain these data, please either upload them as Supporting Information files or deposit them to a stable, public repository and provide us with the relevant URLs, DOIs, or accession numbers. For a list of recommended repositories, please see https://journals.plos.org/plosone/s/recommended-repositories. If there are ethical or legal restrictions on sharing a de-identified data set, please explain them in detail (e.g., data contain potentially sensitive information, data are owned by a third-party organization, etc.) and who has imposed them (e.g., an ethics committee). Please also provide contact information for a data access committee, ethics committee, or other institutional body to which data requests may be sent. If data are owned by a third party, please indicate how others may request data access.

Authors: This manuscript presents a psychometric study, and all relevant data are included within the manuscript. As it is not a bibliographic review or a clinical trial, it is not appropriate to upload the project or data to a repository. The results of the study can be replicated using the data provided in the manuscript.

Authors: The full name of the ethics committee is included in the methods section of the manuscript. It was initially omitted due to confidentiality concerns during peer review. Thank you for your feedback.

Regarding informed consent, the following information is provided in the 'Ethical Aspects' section: “Participants were informed about the purpose of the study and invited to participate. Consent was given verbally and confirmed when participants chose to complete the checklist.”

Authors: The entire reference section has been thoroughly revised. No retracted articles were found, but some typographical errors have been corrected. Additionally, DOIs have been added to all relevant articles. Thank you very much for your feedback.

Additional Editor Comments:

Dear Authors,

Two reviewers have completed their evalation of your manuscript.

Look carefully at their comments and complete your review with the required modifications.

We apologize for the delay in reaching to a decision, and we hope to continue processing your manuscript after minor revisions.

Congratulations for your work and good luck.

Reviewers' comments:

Reviewer's Responses to Questions

Comments to the Author

1. Is the manuscript technically sound, and do the data support the conclusions?

Reviewer #1: Yes

Reviewer #2: Yes

2. Has the statistical analysis been performed appropriately and rigorously?

Reviewer #1: Yes

Reviewer #2: Yes

3. Have the authors made all data underlying the findings in their manuscript fully available?

Reviewer #1: No

Authors: As indicated in the editor's comments, this manuscript is a psychometric study, and all data are included within the manuscript. Since this is not a systematic or scoping review protocol, nor a clinical trial, it is not necessary to upload the project or data to a repository. The study's results can be replicated using the data provided in the manuscript

Reviewer #2: Yes

4. Is the manuscript presented in an intelligible fashion and written in standard English?

Reviewer #1: Yes

Reviewer #2: Yes

5. Review Comments to the Author

Reviewer #1: I would like to congratulate the authors of the manuscript for their work, as I believe that both its subject matter and execution represent a significant contribution to the nursing scientific community, both in clinical and research areas. Therefore, we now make a series of minor observations to ensure that the study meets the quality standards of PLOS ONE.

In general terms, I recommend that the authors review the writing style of the manuscript to ensure compliance with the spelling recommendations in scientific publications.

Introduction

(Lines 109-113) I believe it is necessary to include some information regarding the typical structure of OSCEs, such as the number of scenarios, recommended duration, the context in which they are usually conducted, evaluation forms, etc. The authors recommend that they consult Harden's original reference for this purpose.

Authors: We have added a definition of OSCE, and the paragraph now reads as follows (Lines 109-113):

In simulation, OSCEs are considered the gold standard for competency-based assessments. In OSCEs, candidates rotate through a series of time-limited stations in a circuit designed to assess professional performance in a simulated environment. These assessments facilitate the objective evaluation of a complex range of skills, knowledge, and attitudes through direct observation [4],… 

(Lines 114-116) I think it is relevant to describe what is already known about reliability and agreement in the checklists used in the intensive care unit, to contextualize the need to conduct one concerning an OSCE in this setting.

Authors: We have completed the paragraph with the use of checklists in ICUs. Thank you very much for your suggestion; we believe that the need for our checklist and evaluation through OSCE is now better justified (Lines 116-117).

Methodology

I would recommend that the authors include a figure based on a flow chart detailing each stage of the study.

Authors: We have considered the proposal; however, this is a descriptive study of a metric nature aimed at determining interobserver reliability. The design and content validation process of the NEUMOBACT instrument is detailed in a previous publication:

Reference No. 8 (Raurell-Torredà M, Arrogante O, Aliberch-Raurell AM, Sánchez-Chillón FJ, Torralba-Melero M, Rojo-Rojo A, et al. Design and content validation of a checklist about infection-prevention performance of intensive care nurses in simulation-based scenarios. J Clin Nurs. 2024;33(8):3188-3198. doi:10.1111/jocn.17010) describes the stages of validity and content of the instrument.

(Lines 145-147) The authors mention that the recruitment was carried out at a conference, suggesting convenience sampling. However, this is not explicitly mentioned, which I believe would add transparency to the article.

Authors: We agree with your observation. We have included the type of sampling in the methods section (Line 147).

(Lines 153-159) In addition to mentioning the COSMIN checklist, I think it is important to mention the statistical parameters used and the value at which they were set to calculate the minimum sample size required for the study.

Authors: We have revised the wording of this section to enhance reader comprehension. We did not perform a sample calculation; instead, we followed the COSMIN guidelines for instrument validation.

The updated section now reads as follows (Lines 153-159): “According to the COSMIN Study Design checklist for patient-reported outcome measurement instruments [10], a minimum of 100 pairs of assessments by at least two unbiased evaluators is required to analyze inter-observer reliability. Two independent observers should be used to assess the simulation stations under similar environmental conditions, having received the same instructions. To account for expected losses, a further 10% was added, resulting in a sample of 111 pairs of nurses to evaluate the NEUMOBACT checklist during the simulation”

Reference No. 10: Mokkink LB, Prinsen CA, Patrick DL, Alonso J, Bouter LM, de Vet HCW, et al. COSMIN Study Design checklist for patient-reported outcome measurement instruments. [accessed 5 September 2024]. Available from: https://www.cosmin.nl/wp-content/uploads/COSMIN-study-designing-checklist_final.pdf" 

(Lines 161-168) I would further detail the most relevant characteristics of the simulation scenarios (duration, simulated clinical situation, necessary material and human resources, characteristics of the simulators, etc.).

Authors: Both the simulation scenarios and the methodology, recommended materials, as well as the scripts for carrying out the scenarios are detailed in Supplement 1, which is accompanied by a reference to an article where all cases are fully described. However, following the reviewer’s suggestion, we have added further specifications to provide the reader with a clearer understanding (Lines 168-172).

(Lines 170-172) It is necessary to specify the methodology followed for developing the initial version of NEUMOBACT (it is not mentioned whether it was a literature review, focus groups with thematic analysis and triangulation, etc.).

Authors: The methodology used was the Delphi method. Initially, a group of experts from the Ministry of Health in our country was asked to collaborate using the Delphi method, which is anonymous and conducted in multiple rounds. Subsequently, and in the same way, it was evaluated by clinical simulation experts to assess whether the different simulated actions could be carried out and evaluated (Lines 175-176).

(Lines 202-209) It would be important for the authors to provide the DOI or registration ID of the study in registries such as the Open Science Framework.

Authors: We have considered the suggestion; however, this study is a psychometric study and does not need to be registered in the Open Science Framework. Thank you for the suggestion.

Results

(Lines 235-237) I believe it is relevant to detail the methodology followed for developing the modified version of NEUMOBACT (it is not mentioned whether it was a literature review, focus groups with thematic analysis and triangulation, etc.).

Authors: Thank you for the contribution, you are absolutely right. We have added the technique used to obtain the modified version of NEUMOBACT (Line 239).

Discussion

(Line 278) I would mention future perspectives for upcoming research related to the topic. I would also highlight the potential benefit of applying simulated learning methodologies at the undergraduate and postgraduate levels, such as MAES or OSCEs with validated checklists, to improve patient safety and quality of care.

Authors: Thanks for the suggestion, a few lines have been added in this regard and also a reference that validates it from the literature (Lines 257-260).

Limitations

(Lines 280-288) I believe it is important to incorporate more citations to support the claims made by the authors. I also think they should mention the geographical limitation of NEUMOBACT validation, as well as possible recommendations for its use in contexts outside of Spanish-speaking ones.

Authors: Thank you for the suggestion. We have added a few lines on the matter, (lines 291 -297) along with a reference from the literature to support it:

“It is es

---

## [Decision Letter · Decision Letter 1]

10 Oct 2024

PONE-D-24-06321R1Interrater reliability and agreement of the NEUMOBACT checklist about infection-prevention performance of intensive care nurses in simulation-based scenarios.PLOS ONE

Dear Dr. Zaragoza-García,

Thank you for submitting your manuscript to PLOS ONE. After careful consideration, we feel that it has merit but does not fully meet PLOS ONE’s publication criteria as it currently stands. Therefore, we invite you to submit a revised version of the manuscript that addresses the points raised during the review process.

We look forward to receiving your revised manuscript.

Kind regards,

Javier Fagundo-Rivera, PhD

Academic Editor

PLOS ONE

Journal Requirements:

Additional Editor Comments:

**Dear Authors,**

This round of revisions have complied with the Reviewers comments, and this manuscript could be accepted for publication.

However, before this can be accepted formally, one Reviewer recommends to complete a formal change in the Methodology.

 (Line 144) The authors have added the GRRAS checklist as supplementary material but not completed it.

Please, complete the GRASS checklist with appropriate information, and also your manuscript if necessary, and address it again.

**Best regards.**

Reviewers' comments:

Reviewer's Responses to Questions

**Comments to the Author**

1. If the authors have adequately addressed your comments raised in a previous round of review and you feel that this manuscript is now acceptable for publication, you may indicate that here to bypass the “Comments to the Author” section, enter your conflict of interest statement in the “Confidential to Editor” section, and submit your "Accept" recommendation.

Reviewer #1: All comments have been addressed

Reviewer #2: (No Response)

2. Is the manuscript technically sound, and do the data support the conclusions?

Reviewer #1: Yes

Reviewer #2: Yes

3. Has the statistical analysis been performed appropriately and rigorously? 

Reviewer #1: Yes

Reviewer #2: Yes

4. Have the authors made all data underlying the findings in their manuscript fully available?

Reviewer #1: Yes

Reviewer #2: Yes

5. Is the manuscript presented in an intelligible fashion and written in standard English?

Reviewer #1: Yes

Reviewer #2: Yes

6. Review Comments to the Author

Reviewer #1: I extend my congratulations to the authors for the excellent work they have done, as well as for their efforts in incorporating the suggested recommendations. These adjustments, in my opinion, have significantly improved the overall quality of the article, ensuring that it now aligns more closely with the standards and expectations of the journal. In this regard, I would like to emphasize one final suggestion: I strongly encourage the authors to consider the benefits of registering the studies conducted in dedicated registries designed for this purpose.

Reviewer #2: I would like to thank the authors for their prompt response, as well as for the work of reviewing and implementing most of the suggestions in one way or another. However, I would still like to comment a little note:

Methodology

(Line 144) The authors have added the GRRAS checklist as supplementary material but not completed.

Without further ado, I hope that the article can be accepted soon.

7. PLOS authors have the option to publish the peer review history of their article (what does this mean?). If published, this will include your full peer review and any attached files.

Reviewer #1: No

Reviewer #2: No

---

## [Author Response · Author response to Decision Letter 1]

16 Oct 2024

Authors: Thank you for the opportunity to review the manuscript. We have now addressed the editor's and reviewers' comments. We have added the completed GRRAS checklist. We apologise for not having done so previously.

Journal Requirements:

Authors: They have been rechecked for any errors.

Additional Editor Comments:

Dear Authors,

This round of revisions have complied with the Reviewers comments, and this manuscript could be accepted for publication.

However, before this can be accepted formally, one Reviewer recommends to complete a formal change in the Methodology.

 (Line 144) The authors have added the GRRAS checklist as supplementary material but not completed it.

Please, complete the GRASS checklist with appropriate information, and also your manuscript if necessary, and address it again.

Best regards.

Authors: We apologise for the error. We have attached the completed GRRAS checklist.

Reviewers' comments:

Reviewer's Responses to Questions

Comments to the Author

1. If the authors have adequately addressed your comments raised in a previous round of review and you feel that this manuscript is now acceptable for publication, you may indicate that here to bypass the “Comments to the Author” section, enter your conflict of interest statement in the “Confidential to Editor” section, and submit your "Accept" recommendation.

Reviewer #1: All comments have been addressed

Reviewer #2: (No Response)

Authors: Thank you for your efforts in reviewing our manuscript.

2. Is the manuscript technically sound, and do the data support the conclusions?

Reviewer #1: Yes

Reviewer #2: Yes

3. Has the statistical analysis been performed appropriately and rigorously?

Reviewer #1: Yes

Reviewer #2: Yes

4. Have the authors made all data underlying the findings in their manuscript fully available?

Reviewer #1: Yes

Reviewer #2: Yes

5. Is the manuscript presented in an intelligible fashion and written in standard English?

Reviewer #1: Yes

Reviewer #2: Yes

6. Review Comments to the Author

Reviewer #1: I extend my congratulations to the authors for the excellent work they have done, as well as for their efforts in incorporating the suggested recommendations. These adjustments, in my opinion, have significantly improved the overall quality of the article, ensuring that it now aligns more closely with the standards and expectations of the journal. In this regard, I would like to emphasize one final suggestion: I strongly encourage the authors to consider the benefits of registering the studies conducted in dedicated registries designed for this purpose.

Authors: Thank you for your efforts in reviewing our manuscript. We agree with their recommendations and will take them into account for further studies, which will be registered before being sent for peer review

Reviewer #2: I would like to thank the authors for their prompt response, as well as for the work of reviewing and implementing most of the suggestions in one way or another. However, I would still like to comment a little note:

Methodology

(Line 144) The authors have added the GRRAS checklist as supplementary material but not completed.

Authors: We apologise for the error. We have attached the completed GRRAS checklist.

Without further ado, I hope that the article can be accepted soon.

Authors: Thank you for your efforts in reviewing our manuscript.

7. PLOS authors have the option to publish the peer review history of their article (what does this mean?). If published, this will include your full peer review and any attached files.

Do you want your identity to be public for this peer review? For information about this choice, including consent withdrawal, please see our Privacy Policy.

Reviewer #1: No

Reviewer #2: No

---

## [Editor Report · Decision Letter 2]

21 Oct 2024

Interrater reliability and agreement of the NEUMOBACT checklist about infection-prevention performance of intensive care nurses in simulation-based scenarios.

PONE-D-24-06321R2

Dear Dr. Zaragoza-García,

We’re pleased to inform you that your manuscript has been judged scientifically suitable for publication and will be formally accepted for publication once it meets all outstanding technical requirements.

Kind regards,

Javier Fagundo-Rivera, PhD

Academic Editor

PLOS ONE

Additional Editor Comments (optional):

The authors complied with the Reviewers comments. This manuscript can be accepted now.

---

## [Editor Report · Acceptance letter]

24 Oct 2024

PONE-D-24-06321R2 

PLOS ONE

Dear Dr. Zaragoza-García, 

I'm pleased to inform you that your manuscript has been deemed suitable for publication in PLOS ONE. Congratulations! Your manuscript is now being handed over to our production team.

Kind regards, 

on behalf of

Dr. Javier Fagundo-Rivera 

Academic Editor

PLOS ONE